# Factors Associated with the Evolution of Superficial Vein Thrombosis and Its Impact on the Quality of Life: Results from a Prospective, Unicentric Study

Blanca Ros Gómez [1], Javier Gómez-López [2], Manuel Quintana-Díaz [3], Sheila Victoria Calvo Sevilla [2], Pablo Rodríguez-Fuertes [2], Fabian Tejeda-Jurado [2], Paula Berrocal-Espinosa [2], Juan Francisco Martínez-Ballester [2], Sonia Rodríguez-Roca [2], María Angélica Rivera Núñez [2], Ana M. Martínez Virto [2], Alberto Martín-Vega [4], Carmen Fernández-Capitán [5], Giorgina Salgueiro-Origlia [5], Raquel Marín-Baselga [5], Alicia Lorenzo Hernández [5], Teresa Sancho Bueso [5], Ramón Puchades Rincón de Arellano [5], Belén Gutiérrez-Sancerni [5], Alejandro Díez-Vidal [5], Sergio Carrasco-Molina [5] and Yale Tung-Chen [5,*]

1   Department of Emergency Medicine, Hospital General Universitario Ciudad Real, Obispo Rafael Torija s/n, 13005 Ciudad Real, Spain; blancarosgomez@gmail.com
2   Department of Emergency Medicine, Hospital Universitario La Paz, Paseo Castellana 241, 28046 Madrid, Spain; javigomez1993@hotmail.com (J.G.-L.); victoria_csheila@hotmail.com (S.V.C.S.); pablo.rodriguezfuertes@gmail.com (P.R.-F.); fabian.tejeda@salud.madrid.org (F.T.-J.); paulaberrocal10@gmail.com (P.B.-E.); naugo1@gmail.com (J.F.M.-B.); srroca@salud.madrid.org (S.R.-R.); mangelicariveran@hotmail.com (M.A.R.N.); amvirto@salud.madrid.org (A.M.M.V.)
3   Department of Intensive Care Medicine, Hospital Universitario La Paz, Paseo Castellana 241, 28046 Madrid, Spain; mquintanadiaz@gmail.com
4   Department of Management Control, Hospital Universitario La Paz, Paseo Castellana 241, 28046 Madrid, Spain; amartinv@salud.madrid.org
5   Department of Internal Medicine, Hospital Universitario La Paz, Paseo Castellana 241, 28046 Madrid, Spain; cfdezcapitan@hotmail.com (C.F.-C.); giorgisalgueiro@hotmail.com (G.S.-O.); raquelzgz14@gmail.com (R.M.-B.); alicia.loher@gmail.com (A.L.H.); tsbueso@gmail.com (T.S.B.); ramon.puchades@salud.madrid.org (R.P.R.d.A.); belen.sancerni@gmail.com (B.G.-S.); alejandrodiezv@gmail.com (A.D.-V.); sergiocarrascomolina@gmail.com (S.C.-M.)
*   Correspondence: yale.tung@salud.madrid.org; Tel.: +34-676-030-131

**Abstract:** Background: Superficial venous thrombosis (SVT) is a common clinical condition caused by inflammation and the presence of a thrombus inside a superficial vein. It has traditionally been considered a benign and banal disorder, although it can progress or can be associated with thromboembolic disease of deep territories in up to 20%, asymptomatic or symptomatic pulmonary embolism (PE), especially if it affects the main trunk of the internal saphenous vein. The impact of deep vein thrombosis on the quality of life and its sequelae have long been described in the literature; however, they have not been studied in superficial vein thrombosis. Objectives: We aimed to evaluate the risk factors, management, and complications of SVT and its impact on the quality of life of our patients. Methods: Observational, prospective, single-center study to evaluate the management of SVT. The ultrasound (US) was performed initially on symptomatic patients, during treatment with low-molecular-weight heparin (LMWH), at a follow-up, and at the end of 45 days of treatment. A quality-of-life questionnaire was administered to determine the risk factors, management, and complications of SVT at the moment of diagnosis and at the end of treatment. We included patients referred from the emergency department to a monographic consultation for thromboembolic disease, over 18 years of age with a diagnosis of acute SVT symptomatic, without contraindication to initiate anticoagulation. Results: In total, 63 patients were evaluated between October 2020 and April 2022. The mean age was 65.8 years (SD 13.5), of which 35 were women (55.6%), 39 presented cardiovascular risk factors (61.9%), 25 had a history of previous personal venous thromboembolism (VTE) (39.7%), and 10 had obesity (15.9%), 47 had chronic venous insufficiency or varicose veins (74.9%). During follow-up with ultrasound, 39.7% had partial revascularization, and at discharge, 63.5% had permeabilized the thrombosis against 19% who had residual thrombosis or progression of thrombosis. There was a positive correlation between mobility parameters and improvement in the performance of daily activities (rho = 0.35; *p* = 0.012) and with improvement

in pain/discomfort (rho = 0.37; *p* = 0.007). An improvement in pain parameters was statistically significantly related to a global assessment health perception (rho = 0.48; *p* < 0.001). Anxiety and depression parameters were related to a global assessment health perception (rho = 0.462; *p* = 0.001) and to an overall improvement at 12 months (rho = 0.45; *p* = 0.001). CONCLUSIONS: Superficial venous thrombosis (SVT) is a highly prevalent disease, which is traditionally considered banal and has good evolution, with heterogeneous management in clinical practice and limited information on patient selection for therapies, current treatment routes, and drug use, as well as outcomes. In recent years, the importance of this entity has become evident due to its frequency in clinical practice, its risk of complications, and the impact it has on the quality of life. This study's results emphasize the importance of the diagnosis, treatment, and follow-up of superficial venous thrombosis.

**Keywords:** superficial vein thrombosis; pulmonary embolism; venous thrombosis; anticoagulants; quality of life; outcome

---

## 1. Introduction

Superficial venous thrombosis (SVT), also known as phlebitis or superficial thrombophlebitis, is a clinical condition caused by inflammation and the presence of a thrombus inside a superficial vein.

The most common location is the veins of the lower extremities, mainly affecting the territory of the great saphenous vein [1], and it can present as an inflammatory cord in the territory of the affected superficial vein.

Up to 90% of cases involve varicose veins, since tortuosity predisposes to stasis, inflammation, and thrombosis [2,3]. Other factors to consider are prolonged immobilization, surgery, recent trauma, pregnancy and puerperium, obesity, neoplastic processes, autoimmune diseases, use of contraceptives or hormone replacement therapy, thrombophilias, and personal or family history of venous thromboembolic disease [4].

It is a common pathology in clinical practice with a generally good evolution. SVT has traditionally been considered a benign and banal disorder [5]. However, it can progress or can be associated with thromboembolic disease of deep territories in up to 20% asymptomatic or symptomatic pulmonary embolism (PE). This can especially happen when it affects the main trunk of the internal saphenous vein.

In a meta-analysis [6] that included data from 21 studies, it was observed that 18.1% of patients with SVT had concomitant DVT while 6.9% had concomitant PE, although the study had a high degree of heterogeneity among them.

The impact of deep vein thrombosis on the quality of life and its sequelae have long been described in the literature [7]; however, they have not been studied in superficial vein thrombosis.

We aimed to evaluate the risk factors, management, and complications of SVT and its impact on the quality of life of our patients.

## 2. Materials and Methods

The study was conducted in accordance with the Declaration of Helsinki and was approved by the Ethics Committee of the Hospital Universitario La Paz.

### 2.1. Study Design

In this observational, prospective, and single-center study, the aim was to evaluate the risk factors, management, and complications of SVT that were referred to a thrombosis clinic from the Emergency Department (ED) between October 2020 and April 2022. In total, 63 patients were included in this study.

### 2.2. Patient Selection

All patients who went to the ED and were diagnosed with an acute SVT were recruited and referred to a thrombosis clinic. For this purpose, they had to present an acute symptomatic SVT confirmed by a duplex lower-extremity ultrasound. Those who refused to participate in this study were excluded, as well as patients with contraindication for anticoagulant therapy. All patients were managed with a prophylactic dose of low-weight heparin, unless they had a saphenofemoral or saphenopopliteal junction involvement.

Patients were scheduled for a follow-up at days 15 and 45 from diagnosis to contemplate the presence of early complications and at 3 months to contemplate the presence of late complications. A control ultrasound was performed during treatment with low-molecular-weight heparin and after its completion.

### 2.3. Initial Patient Assessment

Demographic data (age, sex, weight); medical history (comorbidities, medications); risk factors for VTE (previous personal or family history of VTE, pregnancy/puerperium, oral contraceptives, hormone replacement therapy, autoimmune disease, recent surgery (last 3 months), recent immobilization, thrombophilia, malignancy, active or in remission, obesity, venous insufficiency, active chemotherapy); imaging tests performed (duplex lower-extremity ultrasound performed in the ED, during follow-up and prior to discharge; other radiology exams); laboratory tests (creatinine, urea, hemoglobin, hematocrit white blood cells, platelets, D-dimer, INR); variables correlating with therapy (type of anticoagulation, dose, duration); COVID infection or recent vaccination; variables correlating with follow-up (end of symptoms, destination, and date of discharge); recurrence or relapse thrombosis; and variables related to the perception of quality of life EuroQoL-5D Health Questionnaire (EQ-5D).

### 2.4. Ultrasound Data Collection

Radiologists in the department of radiology performed initial whole-leg US examination in patients with the suspicion of having SVT. The follow-up US was performed by an internist from the thrombosis clinic with long-standing experience in vascular US. In both cases, a whole-leg US protocol was performed. The initial study was performed using a Phillips iU22 ultrasound machine with a linear transducer (5–17 MHz) (Phillips España, Madrid, Spain), and the follow-up study was conducted using a GE LOGIQ-e ultrasound system with a linear transducer (5–10 MHz) (General Electrics Healthcare, Madrid, Spain).

### 2.5. Outcome Measures and Definition

The main purpose of this study was to evaluate the management of SVT in clinical practice. The secondary outcome was to evaluate the complications and associated risk factors in patients diagnosed with SVT, as well as the factors associated with the extension, resolution, or recurrence of SVT; to analyze factors associated with residual venous thrombosis; and to determine the impact and evolution in the quality of life of patients with this pathology, allowing for us to identify areas for improvement in the care of patients with SVT disease.

In the EuroQoL-5D Health Questionnaire (EQ-5D), parameters of global assessment of perceived health, mobility, self-care, daily activities, pain/discomfort, anxiety/depression, and global assessment in the last 12 months are determined at inclusion and at discharge of the study.

### 2.6. Statistical Analysis

Baseline characteristics are presented as mean and standard deviation (SD) or median and interquartile range (IQR) for continuous variables and count and proportions for categorical variables. For group comparisons, we used the *t*-test for continuous variables and the Chi-square or Fisher's exact test for categorical variables. Variables with a non-normal distribution were log-transformed before analyzing their differences. To

analyze the relationships between ultrasound evolution and other clinical or analytical parameters, a multivariate regression analysis was performed, in which it was adjusted for predictive variables. Statistical significance was established at $p < 0.05$. Mean values were reported along with 95% confidence intervals computed using bootstrap resampling (1000 repetitions). Statistical significance was set at $p$ value < 0.05. Statistical analyses were conducted with IBM SPSS software v20.0 (SPSS Inc., Chicago, IL, USA).

## 3. Results

The clinical characteristics, risk factors, and comorbidity, as well as concomitant treatment and follow-up, are summarized in Table 1.

**Table 1.** Clinical characteristics of the patients included ($n$ = 63).

| Demographics ($n$ = 63) | $n$ (%) |
|---|---|
| Sex (female)—$n$ (%) | 35 (55.6) |
| Age (years) mean (SD) | 65.84 (13.51) |
| Weight (SD) | 79.23 (14.40) |
| Cardiovascular risk factors (High blood pressure, diabetes Mellitus, dyslipidaemias) (%) | 39 (61.9) |
| Cardiovascular pathology (%) | 5 (7.9) |
| Bronchopathy (%) | 8 (12.7) |
| Autoimmune disease (%) | 7 (11.1) |
| **Laboratory tests at diagnosis—mean (SD)** | |
| Hemoglobin (g/dL) | 14.50 (1.67) |
| White blood cell count ($\times 10^9$/L) | 8.11 (2.40) |
| Platelets ($\times 10^9$/L) | 238.77 (69.66) |
| D-dimer (ng/mL) | 1978.41 (1742.69) |
| INR | 1.00 (0.06) |
| **Risk factors for VTE** | |
| Varicose veins | 47 (74.6) |
| Obesity | 10 (15.9) |
| Neoplasia | 5 (7.9) |
| Active Chemotherapy | 3 (4.8) |
| Previous history of VTE | 25 (39.7) |
| Pregnancy | 1 (1.6) |
| HRT | 1 (1.6) |
| OHC | 1 (1.6) |
| Recent surgery (previous 3 months) | 1 (1.6) |
| Travel > 6 h | 1 (1.6) |
| **Initial therapy** | |
| Enoxaparin | 34 (53.9) |
| Fondaparinux | 18 (28.6) |
| Bemiparin | 11 (17.5) |
| **Extended therapy** | |
| Direct Oral Anticoagulant | 10 (15.9) |
| Acenocumarol | 1 (1.6) |

**Table 1.** *Cont.*

| Demographics (*n* = 63) | *n* (%) |
|---|---|
| **Therapy at discharge** | |
| Sulodexide | 16 (25.4) |
| Acetylsalicylic acid | 6 (9.5) |
| Direct Oral Anticoagulant | 3 (4.8) |
| **Initial ultrasound (*n* = 62)** | |
| GSV | 39 (61.9) |
| SSV | 18 (28.6) |
| GSV + SSV | 2 (3.2) |
| SVT + DVT | 3 (4.8) |
| **Follow-up ultrasound (*n* = 56)** | |
| Recanalization | 9 (14.3) |
| Partially recanalization | 25 (39.7) |
| Thrombosis | 19 (30.2) |
| Progression of thrombosis | 3 (3.2) |
| **Discharge ultrasound (*n* = 58)** | |
| Recanalization | 39 (61.9) |
| Partially recanalization | 8 (12.7) |
| Chronic thrombosis | 8 (12.7) |
| Progression of thrombosis | 3 (4.8) |
| **Recurrence/Relapse** | |
| Recurrence | 11 (17.5) |
| Relapse | 9 (14.3) |
| **COVID-19** | |
| COVID-19 infection | 5 (7.9) |
| Recent vaccination | 7 (11.1) |

VTE: venous thromboembolic disease. SVT: superficial venous thrombosis. DVT: deep vein thrombosis. OHC: oral hormonal contraceptives. HRT: hormone replacement therapy. GSV: great saphenous vein. SSV: small saphenous vein.

A total of 63 patients were evaluated. The average age of the patients was 65.8 years (SD 13.52), of which 35 were women (55.6%), 39 presented cardiovascular risk factors (61.9%), 25 had a previous history of VTE (39.7%), 10 had obesity (15.9%), and 47 had chronic venous insufficiency or varicose veins (74.9%). Meanwhile, 4.8% of the patients had active cancer, and 1.6% underwent surgery in the previous 3 months. Most comorbidities had a low prevalence. The mean D-dimer was 1978.41 (SD 1742.69).

Not all patients had the three ultrasounds performed. There was one patient who did not have the initial ultrasound at inclusion but had it performed during follow-up in the clinic, and seven patients did not have either an ultrasound during follow-up or at discharge.

During follow-up, 39.7% had partial revascularization, and at discharge, 61.9% had recanalized the thrombosis versus the 19% who had a progression of the thrombosis.

Among the risk factors described for thromboembolic disease, a statistically significant association was observed between obesity and the persistence of thrombosis at discharge ($p < 0.05$).

The two patients with thrombus progression during follow-up both had cardiovascular risk factors and were managed, one with an increase from the prophylactic to therapeutic

doses of enoxaparin according to weight while the other was started on rivaroxaban. Of the three patients with thrombus progression before discharge, two were managed with an increase in the enoxaparin dose, and one was started on rivaroxaban.

Likewise, an association close to statistical significance was found between the probability of recurrence and previous history of thromboembolic disease ($p = 0.059$) and the recurrence of another thrombotic episode in female patients ($p = 0.056$).

The quality-of-life parameters of the patients were collected at the moment of inclusion and at discharge (Table 2).

**Table 2.** Quality of life parameters of the patients included at the moment of inclusion and at discharge.

| | Mean at Inclusion (SD) | Mean at Discharge (SD) | *p*-Value |
|---|---|---|---|
| Global health perception (1, worst—10, no problem) | 6.21 (1.74) | 7.19 (1.68) | <0.001 |
| Mobility (1, no problem—3, severe problem) | 1.56 (0.53) | 1.19 (0.39) | <0.001 |
| Personal care (1, no problem—3, severe problem) | 1.27 (0.44) | 1.04 (0.19) | 0.001 |
| Daily activities (1, no problem—3, severe problem) | 1.27 (0.44) | 1.04 (0.19) | <0.001 |
| Pain/discomfort (1, no problem—3, severe problem) | 2.33 (0.64) | 1.58 (0.63) | <0.001 |
| Anxiety/depression (1, no problem—3, severe problem) | 1.75 (0.73) | 1.25 (0.48) | <0.001 |
| Last 12 months (1, no problem—3, severe problem) | 2.44 (0.63) | 1.48 (0.57) | <0.001 |

There was a positive correlation between mobility parameters and improvement in the performance of daily activities (rho = 0.37; $p = 0.012$) and with improvement in pain/discomfort (rho = 0.37; $p = 0.007$).

An improvement in pain parameters was statistically significantly related to global assessment health perception (rho = 0.48; $p < 0.001$).

Anxiety and depression parameters were related to a global assessment health perception (rho = 0.46; $p = 0.001$) and to an overall improvement at 12 months (rho = 0.45; $p = 0.001$).

We did not find significant associations between age, sex, weight, obesity, thrombosis recurrence or progression, and duration of symptoms with global assessment at inclusion or discharge. Only the absence of cardiovascular risk factors was correlated with a better global perception at discharge (Rho = −0.38; $p = 0.025$).

## 4. Discussion

Superficial Vein Thrombosis (SVT) can be classified into SVT on varicose veins and SVT on non-varicose veins, which includes different disorders where thrombosis and inflammation with intimal proliferation and fibrosis of the media variably predominate.

The most frequent form of superficial thrombosis in the lower limbs is in the varicose veins, occurring in up to 90% of cases, since tortuosity predisposes to stasis, inflammation, and thrombosis [2,3]. Other factors to take into account are prolonged immobilization, surgery, recent trauma, pregnancy and puerperium, obesity, neoplastic processes, autoimmune diseases, use of contraceptives or hormone replacement therapy, thrombophilias, and personal or family history of venous thromboembolic disease [4].

The characteristics of the patients included in our study were similar to those of previous studies: the average age was 65 years, diagnoses were predominant in women,

and there was a high percentage of patients with varicose veins and a history of VTE. Only the presence of obesity was lower than that observed in other studies (15.9% compared to 86.3% in the POST study [8]), probably due to underestimation, since BMI is seldom recorded in the history.

Generally, diagnosis is based on clinical suspicion and the use of an ultrasound Doppler to confirm the diagnosis.

While the role of D-dimer determination for the diagnosis of DVT is widely reported in the literature, in the case of SVT, it is less studied. A 2015 study [9] found that the sensitivity and specificity of D-dimer for DVT were 92% and 60%, respectively, with a negative predictive value of 98%, while for SVT, they were 77% and 60%, respectively, with a negative predictive value of 93%.

The ultrasound Doppler is the gold standard in the diagnosis of SVT, since it is a simple, reproducible, non-invasive, and inexpensive technique. The absence of collapse after compression with the probe of the contiguous tissues is the main ultrasound finding and the only one that allows for establishing the diagnosis of venous thrombosis [10].

Another ultrasound sign to be evaluated is the presence of blood flow within the vessel as seen by an ultrasound Doppler with distal compression of the limb. Under normal conditions, increased flow is observed. The absence of or decrease in flow in the ultrasound Doppler may be related to the presence of intramural thrombosis. However, it should not be used as the sole diagnostic criterion due to its low sensitivity [11].

In thrombosis, the diameter of the vessel is usually increased in relation to the normal caliber, although this finding is not specific, as well as the presence of intraluminal thrombus [12].

During the follow-up, the patients should be re-evaluated 7–10 days after diagnosis to assess the evolution of SVT, and a Doppler ultrasound should be performed 8–15 days after starting treatment. In case of clinical progression, it is recommended to perform a control ultrasound to exclude extension to the deep venous system [6].

However, the management of SVT is characterized by being very heterogeneous, since the indications for treatment have changed in recent years as it has been discovered that SVT is not such a trivial disease.

Standard clinical practice includes a wide spectrum of therapeutic approaches (including oral and injectable anticoagulants, non-steroidal anti-inflammatory drugs (NSAID), non-drug therapy, compression therapy, and no-treatment) [13].

The CALISTO study [14], with a total of 3002 patients, compared the treatment, for 45 days, of isolated lower-limb DVT of at least 5 cm in length diagnosed using US Doppler with fondaparinux at a dose of 2.5 mg subcutaneously every 24 h versus placebo, finding an 85% reduction in the risk of symptomatic thromboembolic complications, as well as a reduction in the extent and recurrence of DVT. The incidence of serious adverse events was 0.7% versus 1.1% for placebo.

In the STENOX study [15], 427 patients were randomized to four treatment groups: low-molecular-weight heparine (LMWH) at prophylactic doses, LMWH at therapeutic doses, NSAID (Tenoxicam 20 mg/day), and placebo. The occurrence of VTE was evaluated, and results were analyzed. The incidence of VTE appeared to be lower in LMWH treatment compared to placebo, although it was not statistically significant. Both doses of LMWH significantly reduced the extent and/or recurrence of SVT compared to placebo. However, the combined end points of VTE recurrent SVT and the extension of SVT were significantly reduced in all active treatment groups compared to placebo.

For all these reasons, a Cochrane review [13] reports the association of fondaparinux with a significantly lower incidence of VTE, extension of SVT, or recurrence of SVT compared to placebo with a similar bleeding risk. The use of rivaroxaban in this setting requires further study. Compared with placebo, LMWH and NSAIDs appeared to reduce the extent and recurrence of SVT.

INSIGHTS-SVT [16] was a large prospective observational study on the current treatment of SVT, providing information on SVT patient characteristics, diagnosis, management,

and outcomes. The study showed that the risk profiles, clinical presentation, and treatment patterns of patients were very heterogeneous. Despite a high rate of initial anticoagulation therapy, the risk of recurrent vascular complications in patients was remarkably high.

The study showed a high risk of thromboembolic complications in the management of acute isolated DVT in real life despite antithrombotic treatment, and 7.4% of patients did not improve or became worse after three months. Importantly, the study identified a history of previous DVT and thrombus length as independent factors that increased the risk of recurrent VTE.

In addition, during follow-up, it was detected that almost half of the patients in the study (48.6%) did not receive anticoagulation therapy beyond 25 days, indicating a gap in adherence between guideline recommendation and clinical practice.

Therefore, it seems reasonable to think that improving the staging of patients with SVT and improving adherence to treatment and follow-up may reduce the complications associated with the evolution of SVT.

The goals of the DVT treatment include reducing symptoms, preventing the extension of thrombosis including progression to the deep venous system, avoiding recurrence, and preventing thromboembolic complications, mainly for DVT and PE [17].

Quality of life is increasingly seen as an important outcome in clinical care. Etiology, diagnosis, and management of venous thrombosis have been studied extensively, but only a few studies have examined the impact of venous thrombosis on the quality of life [18]. This is the first study to use the EQ-5D to compare the disutility in SVT patients with that of other chronic diseases. Consistent with previous estimates [19], our results confirm and emphasize the significant disutility associated with this disease. SVT patients suffer from the same deterioration as patients with other serious chronic diseases, including cancer and cardiovascular diseases. The results indicate that SVT has a significant physical, social, and psychological impact, and is, therefore, a must-treat disease.

Post-thrombotic syndrome is one of the main sequelae of thrombosis, with a high prevalence that can significantly limit the patient's quality of life [20]. The symptoms are highly variable but are frequently associated with pain in the extremities, sensation of heaviness, edema, cramps, and hyperpigmentation due to stasis and ulcers in the most severe cases. It usually worsens with standing and improves with decubitus and an elevation of the extremities [21].

A strength of our study is that all patients had to go through at least 2–3 Doppler ultrasound exams, which allowed for us to confirm the diagnosis of SVT and stage the severity of the disease (thrombus length, DVT involvement, ...). The second strength is in the longitudinal study design and the high rate of available EuroQoL-5D Health Questionnaire (EQ-5D) data (>90% of study participants). Furthermore, using this well-validated psychometric instrument can also be considered one of our study's strengths.

Our study has several limitations. The management of SVT in routine clinical practice is very heterogeneous; on many occasions, complementary tests and hospital referrals are not performed, driven by the idea that it is a banal pathology. Another limitation is that the initial ultrasound and the follow-up ultrasound were performed by different physicians, on different machines, and under different conditions. Although the impact of this limitation is small because both groups had a similar experience, the same protocol was used, and the machines were of similar quality. The small number of subjects and the single-center study design would limit external validity. Therefore, we believe that this study could be useful for the design of future multicenter studies, which could change our clinical practice.

## 5. Conclusions

Superficial venous thrombosis has traditionally been considered a banal and benign entity, so its management in clinical practice has been highly varied.

In recent years, the importance of this entity has become evident due to its frequency in clinical practice, its risk of complications, and the impact it has on the quality of life.

SVT patients showed significant improvements in EuroQoL-5D and reduced anxiety, as well as discomfort during the follow-up and improved global health perception. We could demonstrate that the range of disutility among SVT patients was within the ranges of other chronic diseases (cardiovascular, end-stage renal, liver, cancer, and visual diseases), and EuroQoL-5D is an appropriate tool to monitor change.

This study's results emphasize the importance of the diagnosis, treatment, and follow-up of superficial venous thrombosis.

**Author Contributions:** All authors have contributed to this work. Conception and design: B.R.G., M.Q.-D. and Y.T.-C.; Analysis and interpretation: B.R.G., J.G.-L., P.R.-F., F.T.-J., P.B.-E., J.F.M.-B., S.R.-R., M.A.R.N., A.M.M.V., M.Q.-D., A.M.-V., Y.T.-C., C.F.-C., G.S.-O., A.L.H., T.S.B., R.P.R.d.A. and S.V.C.S.; Data collection: B.R.G., J.G.-L., P.R.-F., F.T.-J., P.B.-E., J.F.M.-B., S.R.-R. and S.V.C.S.; Writing the article: B.R.G. and Y.T.-C.; Critical revision of the article: B.R.G., J.G.-L., P.R.-F., F.T.-J., P.B.-E., J.F.M.-B., S.R.-R., M.A.R.N., A.M.M.V., M.Q.-D., A.M.-V., Y.T.-C., C.F.-C., G.S.-O., A.L.H., T.S.B., R.P.R.d.A., B.G.-S., A.D.-V. and S.C.-M.; Final approval of the article: B.R.G., J.G.-L., P.R.-F., F.T.-J., P.B.-E., J.F.M.-B., S.R.-R., M.A.R.N., A.M.M.V., M.Q.-D., A.M.-V., Y.T.-C., C.F.-C., G.S.-O., A.L.H., T.S.B., R.P.R.d.A., S.V.C.S., B.G.-S., A.D.-V. and S.C.-M.; Statistical analysis: R.M.-B. and Y.T.-C.; Overall responsibility: B.R.G. and Y.T.-C. All authors have read and agreed to the published version of the manuscript.

**Funding:** This research was funded by grant from the Madrid-Castilla La Mancha Society of Internal Medicine (SOMIMACA). There was no additional external funding received for this study.

**Institutional Review Board Statement:** The study was conducted according to the guidelines of the Declaration of Helsinki. The study was approved by the Local Institutional Review Board (H. U. La Paz—PI3870).

**Informed Consent Statement:** Informed consent was obtained from all subjects involved in the study.

**Data Availability Statement:** The authors confirm that the data supporting the findings of this study are available from the corresponding author upon reasonable request.

**Conflicts of Interest:** The authors declare no conflicts of interest.

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
