# Peer review of "Factors Associated with the Evolution of Superficial Vein Thrombosis and Its Impact on the Quality of Life: Results from a Prospective, Unicentric Study"

_2813-2475, doi:10.3390/jvd3010001_

Round 1
Reviewer 1 Report
Comments and Suggestions for Authors
The authors in their manuscript (MS) entitled " Factors associated with the evolution of superficial vein thrombosis and its impact on the quality of life: results from a prospective, unicentric study" provide prospective analysis of the management of superficial vein thrombosis.
The MS is generally well written and reference vast literature regarding the subject. The provided statistical analysis is correctly done and the data is clearly presented.
The topic of SVT which authors chose is much less studied than deep vein thrombosis and still there's no clear consensus what should be the exact treatment of this pathology.
Discussion touches many issues associated with the problem and references many publications from the field.
Major points:
1. A paragraph could be added in the discussion or conclusions to clearly summarize the results.
2. The manuscript requires English language corrections. I suggest shortening the sentences (or dividing them into 2 separate). In current version the sentences are very long and it's hard to understand their meaning.
Ex:
"It is a common pathology in clinical practice, generally with a good evolution, which 78 is why SVT has traditionally been considered a benign and banal disorder5, although it can progress or can be associated with thromboembolic disease of deep territories in up 80 to 20%, asymptomatic or symptomatic pulmonary embolism (PE), especially if it affects 81 the main trunk of the internal saphenous vein."-
My proposition
It is a common pathology in clinical practice with a generally good evolution. SVT has traditionally been considered a benign and banal disorder. However, it can progress or can be associated with thromboembolic disease of deep territories in up to 20%, asymptomatic or symptomatic pulmonary embolism (PE). This can especially happen when it affects the main trunk of the internal saphenous vein.
Minor points:
1. Some small editing mistakes:
a) Among the risk factors described for thromboembolic disease, a statistically signifi- 164 cant association was observed between obesity and persistence of thrombosis at discharge 165 (p< 0,05).
b) I would keep same length in SD. For example in Abstract :
The mean age was 65.84 years (SD 13,518), of which 35 were women 45
later on: A total of 63 patients were evaluated. The mean age was 65,8 years (SD 13,52), of 158
Comments on the Quality of English Language
The manuscript requires English language corrections. I suggest shortening the sentences (or dividing them into 2 separate). In current version the sentences are very long and it's hard to understand their meaning.
Ex:
"It is a common pathology in clinical practice, generally with a good evolution, which 78 is why SVT has traditionally been considered a benign and banal disorder5, although it can progress or can be associated with thromboembolic disease of deep territories in up 80 to 20%, asymptomatic or symptomatic pulmonary embolism (PE), especially if it affects 81 the main trunk of the internal saphenous vein."-
My proposition
It is a common pathology in clinical practice with a generally good evolution. SVT has traditionally been considered a benign and banal disorder. However, it can progress or can be associated with thromboembolic disease of deep territories in up to 20%, asymptomatic or symptomatic pulmonary embolism (PE). This can especially happen when it affects the main trunk of the internal saphenous vein.
Author Response
Reviewer 1:
Comments and Suggestions for Authors
The authors in their manuscript (MS) entitled "Factors associated with the evolution of superficial vein thrombosis and its impact on the quality of life: results from a prospective, unicentric study" provide prospective analysis of the management of superficial vein thrombosis.
The MS is generally well written and reference vast literature regarding the subject. The provided statistical analysis is correctly done and the data is clearly presented.
The topic of SVT which authors chose is much less studied than deep vein thrombosis and still there's no clear consensus what should be the exact treatment of this pathology.
Discussion touches many issues associated with the problem and references many publications from the field.
Major points:
- A paragraph could be added in the discussion or conclusions to clearly summarize the results.
RESPONSE: We thank the reviewer for the helpful comment. We have amended the discussion as follows:
“Superficial venous thrombosis has traditionally been considered a banal and benign entity, so its management in clinical practice has been very varied.
In recent years, the importance of this entity has become evident due to its frequency in clinical practice, its risk of complications and the impact it has on quality of life.
SVT patients showed significant improvement in EuroQoL‑5D and reduced anxiety, discomfort during the follow-up among improved global health perception. We could demonstrate that the range of disutility among SVT patients are within the ranges of other chronic diseases (cardiovascular, end-stage renal, liver, cancer, and visual diseases) EuroQoL‑5D is an appropriate tool to monitor change.
This study results emphasize the importance of the diagnosis, treatment, and follow-up of superficial venous thrombosis.”
Also added to conclusion:
“Quality of life is increasingly seen as an important outcome in clinical care. Etiology, diagnosis, and management of venous thrombosis have been studied extensively, but only few studies have examined the impact of venous thrombosis on quality of life [18]. This is the first study to use the EQ-5D to compare the disutility in SVT patients with that of other chronic diseases. Consistent with previous estimates [19] our results confirm and emphasize the significant disutility associated with this disease. SVT patients suffer from the same deterioration as patients with other serious chronic diseases including cancer and cardiovascular diseases. The results indicate that SVT has a significant physical, social, and psychological impact, and is therefore a must-treat disease.
Post-thrombotic syndrome is one of the main sequelae of thrombosis, with a high prevalence that can significantly limit the patient's quality of life [20]. The symptoms are highly variable, but are frequently associated with pain in the extremities, sensation of heaviness, edema, cramps, hyperpigmentation due to stasis and ulcers in the most severe cases. It usually worsens with standing and improves with decubitus and elevation of the extremities [21].
A strength of our study is that all patients had to go through at least 2-3 Doppler ultrasound exams, which allowed us to confirm the diagnosis of SVT and stage the severity of the disease (thrombus length, DVT involvement,…). Secondly, the longitudinal study design and the high rate of available EuroQoL-5D Health Questionnaire (EQ-5D) data (> 90% of study participants). Furthermore, using this well-validated psychometric instrument can be considered one of our study’s strengths.”
- The manuscript requires English language corrections. I suggest shortening the sentences (or dividing them into 2 separate). In current version the sentences are very long and it's hard to understand their meaning.
Ex:
"It is a common pathology in clinical practice, generally with a good evolution, which 78 is why SVT has traditionally been considered a benign and banal disorder5, although it can progress or can be associated with thromboembolic disease of deep territories in up 80 to 20%, asymptomatic or symptomatic pulmonary embolism (PE), especially if it affects 81 the main trunk of the internal saphenous vein."-
My proposition
It is a common pathology in clinical practice with a generally good evolution. SVT has traditionally been considered a benign and banal disorder. However, it can progress or can be associated with thromboembolic disease of deep territories in up to 20%, asymptomatic or symptomatic pulmonary embolism (PE). This can especially happen when it affects the main trunk of the internal saphenous vein.
RESPONSE: We greatly appreciate reviewer suggestion. We revised the text for any grammatical, linguistic, or spelling mistakes. We corrected and shortened different sentences in the text.
Minor points:
- Some small editing mistakes:
- a) Among the risk factors described for thromboembolic disease, a statistically signifi- 164 cant association was observed between obesity and persistence of thrombosis at discharge 165 (p< 0,05).
- b) I would keep same length in SD. For example in Abstract :
The mean age was 65.84 years (SD 13,518), of which 35 were women 45
later on: A total of 63 patients were evaluated. The mean age was 65,8 years (SD 13,52), of 158
RESPONSE: We greatly appreciate reviewer suggestion. We revised the text for any grammatical, linguistic, or spelling mistakes. We corrected some typing mistakes and grammatical errors as the ones pointed out.
Reviewer 2 Report
Comments and Suggestions for Authors
Authors have described a small setting of patients presenting with superficial vein thrombosis and investigated its impact on the quality of life.
The manuscript is interesting but contains a series of flaws and needs to be gratly ameliorated.
First at all, patients should be better described because it seems that a portion of them already experienced a venous thromboembolism.
Then, Authors should check tables beacuse numbers did not sum sometimes to 63 (or 100%).
Authors should discuss results obtained in comparison to known data and not only comment previous data.
Author Response
Reviewer 2:
Comments and Suggestions for Authors
Authors have described a small setting of patients presenting with superficial vein thrombosis and investigated its impact on the quality of life.
The manuscript is interesting but contains a series of flaws and needs to be gratly ameliorated.
First at all, patients should be better described because it seems that a portion of them already experienced a venous thromboembolism.
RESPONSE: we thank the reviewer’s suggestion. We amended the following paragraph in the results section:
“The average age of the patients was 65,8 years (SD 13,52), of which 35 were women (55,6%), 39 presented cardiovascular risk factors (61,9%), 25 had a previous history of VTE (39,7%), 10 had obesity (15,9%), 47 had chronic venous insufficiency or varicose veins (74,9%). 4.8% of patients had active cancer and 1,6% underwent surgery in the previous 3 months. Most comorbidities had a low prevalence. The mean D-dimer was 1978,41 (SD 1742,69).”
Then, Authors should check tables beacuse numbers did not sum sometimes to 63 (or 100%).
RESPONSE: we thank the reviewer helpful comment. We have double checked the tables. We agree that many variables do not sum up 63. To be more specific, the parameters of “initial ultrasound” (62), “follow-up ultrasound” (56), “discharge ultrasound” (58). This is due to missing values, since not all patients had the 3 ultrasounds performed due to different reasons.
Authors should discuss results obtained in comparison to known data and not only comment previous data.
RESPONSE: We thank the reviewer for this helpful advice. Following the suggestion, we added the following paragraph: “The characteristics of the patients included in our study were similar to previous studies: the average age was 65 years, a predominance of diagnoses in women, and a high percentage of patients with varicose veins and a history of VTE. Only the presence of obesity was lower than that observed in other studies (15,9% compared to 86.3% in the POST study), probably due to an underestimation, since BMI is seldomly recorded in the history.”
Reviewer 3 Report
Comments and Suggestions for Authors
The manuscript is written well, the methodology, results and conclusions are properly shaped. Please complete the management part of discussions with interventional aspects, compressions stockings and long term antithrombotic management strategies.
Comments on the Quality of English Language
has to be improved
Author Response
Reviewer 3:
Comments and Suggestions for Authors
The manuscript is written well, the methodology, results and conclusions are properly shaped. Please complete the management part of discussions with interventional aspects, compressions stockings and long term antithrombotic management strategies.
RESPONSE: we thank the reviewer suggestion. We have added to previously described initial and extended therapy a new section about therapy at discharge. Unfortunately we did not record the use nor adherence to compression stockings.
Therapy at discharge |
|
Sulodexide |
16 (25,4) |
Acetylsalicylic acid |
6 (9,5) |
Direct Oral Anticoagulant |
3 (4,8) |
Comments on the Quality of English Language
has to be improved
RESPONSE: We thank the reviewer for the helpful comment. We revised the text for any grammatical, linguistic or spelling mistakes. We corrected some typing mistakes and grammatical errors.
Round 2
Reviewer 2 Report
Comments and Suggestions for Authors
Authors have implemented their work.
I suggest to specify in the legend of tables why some totals are not 63 (i.e . Follow-up ultrasound: 56. Discharge ultrasound: 58). Patients were lost at the follow-up? or what?
Author Response
Authors have implemented their work.
I suggest to specify in the legend of tables why some totals are not 63 (i.e . Follow-up ultrasound: 56. Discharge ultrasound: 58). Patients were lost at the follow-up? or what?
RESPONSE: we thank the reviewer for your comment. It is difficult to know the exact reason, as it might be that the patient missed the appointment or that the follow-up ultrasound was not performed during the second visit, waiting for the discharge ultrasound, even more, that the patient refused to have the exam. We have added the N values to each ultrasound, and the following sentence in the text:
“Not all patients had the three ultrasounds performed. There was one patient who did not have the initial ultrasound at inclusion, but performed during follow-up in the clinic, and seven patients did not have either an ultrasound during follow-up or discharge.”